# Hepatic Differentiation of Stem Cells in 2D and 3D Biomaterial Systems

**DOI:** 10.3390/bioengineering7020047

**Published:** 2020-05-25

**Authors:** Xiaoyu Zhao, Yanlun Zhu, Andrew L. Laslett, Hon Fai Chan

**Affiliations:** 1Institute for Tissue Engineering and Regenerative Medicine, The Chinese University of Hong Kong, Hong Kong 999077, China; xiaoyuzhao@cuhk.edu.hk (X.Z.); 1155136524@link.cuhk.edu.hk (Y.Z.); 2Key Laboratory for Regenerative Medicine, Ministry of Education, School of Biomedical Sciences, Faculty of Medicine, The Chinese University of Hong Kong, Hong Kong 999077, China; 3CSIRO Manufacturing, Clayton, Victoria 3168, Australia; Andrew.Laslett@csiro.au; 4Australian Regenerative Medicine Institute, Monash University, Victoria 3800, Australia

**Keywords:** hepatic differentiation, stem cell, biomaterial, decellularized extracellular matrix, stiffness, topography

## Abstract

A critical shortage of donor livers for treating end-stage liver failure signifies the urgent need for alternative treatment options. Hepatocyte-like cells (HLC) derived from various stem cells represent a promising cell source for hepatocyte transplantation, liver tissue engineering, and development of a bioartificial liver assist device. At present, the protocols of hepatic differentiation of stem cells are optimized based on soluble chemical signals introduced in the culture medium and the HLC produced typically retain an immature phenotype. To promote further hepatic differentiation and maturation, biomaterials can be designed to recapitulate cell–extracellular matrix (ECM) interactions in both 2D and 3D configurations. In this review, we will summarize and compare various 2D and 3D biomaterial systems that have been applied to hepatic differentiation, and highlight their roles in presenting biochemical and physical cues to different stem cell sources.

## 1. Introduction

Liver, the largest internal organ in our body, performs many important functions including protein synthesis, detoxification, metabolism and bile secretion. The liver has a remarkable capability to regenerate after injury or resection, such as after partial hepatectomy. Nevertheless, in situations such as acute liver injury or end-stage liver disease, liver regeneration is insufficient resulting in liver failure and eventually death [1]. Liver disease accounts for approximately 2 million deaths per year worldwide [2]. The leading causes of liver-associated deaths include liver cirrhosis, viral hepatitis, and hepatocellular carcinoma. In particular, liver cirrhosis and liver cancer are currently the 11th and 16th most common causes of death globally [2]. Together, they account for 3.5% of all deaths worldwide. Liver transplantation, as the only definite treatment for acute liver failure and end-stage liver disease, is hampered by the limited supply of donor organs [3]. Consequently, alternative treatments are desperately needed to combat severe liver diseases.

Hepatocyte transplantation and tissue engineering are deemed as promising alternatives to liver transplantation. Transplantation of hepatocytes instead of intact liver is advantageous as it is a less invasive procedure and permits the use of cryopreserved cells [4]. On the other hand, tissue engineering combines both cells and scaffolds to develop biological substitutes to restore or replace damaged tissues or organs, and has been used to reconstruct various tissues/organs such as skin, liver, spinal cord and blood vessels for implantation [5]. Hepatocytes and engineered liver constructs can also be incorporated in bioartificial liver assist devices to offer temporary support to liver functions [6]. Central to these approaches is the requirement of a sustainable cell source which cannot be met by primary hepatocytes due to a shortage of donor livers. To overcome the shortage of hepatocytes, scientists are actively pursuing the derivation of functional hepatocytes from stem cells, including mesenchymal stem cells (MSC), embryonic stem cells (ESC), induced pluripotent stem cells (iPSC), and hepatic progenitor/stem cells (HPC) [7]. Stem cells are an attractive cell source, characterized by a self-renewal capacity as well as potential to differentiate into diverse cell progenies, including the hepatic lineage. Therefore, hepatocyte-like cells (HLC) differentiated from stem cells, with morphological, phenotypic, and functional characteristics of mature hepatocytes, could potentially be employed in hepatocyte transplantation and liver tissue engineering. 

Typically, stem cells are differentiated by introducing various combinations of soluble chemical signals (e.g., growth factors or small molecules) to coax the cells into becoming HLC, usually *via* a stepwise strategy in 2D or 3D [8]. Monolayer culture is the most common method to induce differentiation, but 3D differentiation involving culturing stem cells such as embryoid bodies or spheroids and encapsulating cells in a scaffold has also been reported [9]. Although a number of reports have attempted to optimize the differentiation protocol in terms of differentiation efficiency and cost, the HLC produced are mostly immature in nature (i.e., expressing early hepatic markers and low levels of mature hepatic markers and cytochrome P450 (CYP) activities) and cannot maintain a long-term differentiated phenotype [10,11,12]. This has led to the question of whether additional cues should be supplied in order to further improve the differentiation process. 

Previous studies have shown that the *in vivo* environment can provide necessary signals to foster the maturation of stem cell-derived terminal cell types [13]. The *in vivo* cellular microenvironment contains not only soluble factors but also stromal cells and an insoluble extracellular matrix (ECM), a complex and dynamic network of macromolecules [14]. The hepatic differentiation efficiency of stem cells such as ESC has been shown to increase in the presence of stromal cells such as endothelial cells and fibroblasts [15,16]. By mixing hepatic progenitor cells with mesenchymal stem cells and/or endothelial cells, liver bud organoids could be produced with an expression profile more similar to human hepatic tissue [17]. 

In term of insoluble ECM, examples of macromolecules include collagen, fibronectin, and hyaluronic acid. Although the ECM was once considered an inert supportive scaffold, it is becoming evident that ECM plays an important role in organ development, function and wound repair [18,19,20]. The relative amounts and organization of different ECM components differ for each tissue which give rise to distinct physical and biochemical ECM properties [21]. Physical properties such as stiffness, porosity and topography have been shown to modulate stem cell differentiation [22,23,24]. From a biochemical point of view, the ECM components can regulate stem cell differentiation *via* direct binding with specific cell surface integrin (e.g., α5β1) or non-canonical growth factor presentation [25,26]. Therefore, natural or synthetic biomaterials can be employed to present physical and biochemical cues, which serve as additional stimuli to enhance hepatic differentiation of stem cells. While biomaterials such as collagen, laminin and decellularized ECM are traditionally applied as a 2D coating, the development of 3D biomaterial scaffolds has provided an alternative to influence cell fate *via* supplying ECM cues in 3D. Therefore, a systematic overview comparing and contrasting 2D and 3D biomaterial systems and their effects on stem cells is timely, providing a platform for future design of biomaterials to efficiently induce hepatic differentiation of stem cells for clinical and commercial applications.

In this review, we will discuss the application of various biomaterial systems in hepatic differentiation of stem cells in 2D and 3D culture (Figure 1), highlighting their role in promoting lineage specification and hepatic maturation. We start with introducing various stem cell sources, and then summarizing the examples of hepatic differentiation in 2D and 3D biomaterial systems. Finally, we will compare and contrast the effects of ECM on 2D and 3D culture. 

## 2. Sources of Stem Cells

The inherent limitations of primary hepatocytes, including limited supply and rapid dedifferentiation during *in vitro* culture, have spurred efforts to explore alternative cell sources, including ESC, iPSC, MSC, HPC. ESC are derived from the inner cell mass of the blastocyst, are expandable, and can differentiate into three germ layer cell types [27]. The controversy over the ethical issues surrounding the procurement of ESC has been circumvented by the discovery of iPSC, which can be generated from terminally differentiated adult cells by forcibly expressing a selected group of transcription factors in the cells [28]. Both ESC and iPSC are pluripotent and can be directed to differentiate into HLC *via* a similar process paralleling the sequential hepatic development *in vivo* [29,30,31,32,33,34]. In one example, the differentiation process involves first inducing ESC and iPSC to differentiate into the definitive endoderm using growth factors or supplements such as Activin A and B27. Then hepatic endoderm is specified by addition of growth factors such as BMP-4 and FGF-2. This is followed by differentiation into immature and mature hepatocytes *via* adding HGF and oncostatin M, respectively [30]. Nevertheless, the potential of teratoma formation after implantation of both cell types poses a major risk of clinical use. 

MSC, on the other hand, can be derived from different tissues, such as bone marrow, adipose tissue, placenta [35]. MSC are known to be multipotent and not pluripotent, so they do not contribute to teratoma formation [36]. In addition to differentiating into mesenchyme-related lineages, MSC can also transdifferentiate into HLC in the presence of a specialized array of growth factors [37]. A two-step protocol was reported in one example, in which MSC was first treated with HGF, bFGF, and nicotinamide. The supplementation of oncostatin M, dexamethasone, and ITS was used to stimulate maturation thereafter [37].

Finally, HPC, also termed as oval cells, can be isolated and expanded from donated livers unsuitable for transplantation, which can then be induced to differentiate between hepatocyte and biliary lineages [38]. To induce differentiation to hepatocytes specifically, HPC can be cultured in the presence of HGF and FGF9 [39]. Meanwhile, human HPC cell lines, such as HepaRG, are also available for studying hepatic differentiation [40], and a similar stem cell population, termed resident liver stem cells and different from hepatic progenitor/stem cells by not expressing albumin, have also been reported [41]. One of the disadvantages, however, is the shortage of donor liver for HPC or liver stem cell isolation. Overall, a number of stem cell sources may be applied to produce HLC for clinical use for the treatment of liver diseases.

To determine differentiation efficiency, the expression of markers of various stages of differentiation can be evaluated, including Foxa2, Sox17 (definite endoderm), Hnf4α (hepatic endoderm), AFP (immature hepatocyte) and/or albumin (mature hepatocyte) [30]. In addition, the secretion of albumin and urea can serve as functional outputs of HLC. HLC should also be able to storage glycogen and lipids. Finally, the metabolic functions of HLC can be assessed by measuring the expression and activity of enzymes such as CYP450 [31]. Next, we will discuss how various 2D and 3D biomaterial systems modulate hepatic differentiation of stem cells.

## 3. Biomaterial Systems Employed in Hepatic Differentiation of Stem Cells 

Biomaterial was once defined as “a nonviable material used in a medical device, intended to interact with biological systems” as it was considered an inert supportive scaffold [42]. The definition was later revised as “material intended to interface with biological systems to evaluate, treat, augment or replace any tissue, organ or function of the body”, reflecting an influencing the role of biomaterials on the human body in the context of tissue regeneration [43]. Biomaterials for modulating stem cell differentiation can be generally categorized into natural and synthetic polymers [44], and can be applied as 2D coatings or 3D scaffolds [45,46]. 3D scaffolds can be fabricated with conventional methods such as freeze-drying, particle leaching, and gas foam, whereas more advanced processing technologies such as 3D printing and microfluidic-based cell encapsulation have been recently developed to facilitate the control of scaffold properties (A detailed review of the fabrication technologies for tissue engineering was published elsewhere [47,48,49]). 

Naturally-derived polymers, including protein and polysaccharides, have the potential advantage of biological recognition that renders them bioactive. However, natural materials often bring concerns such as weak mechanical strength and quick degradation rates [50]. Currently, Matrigel (or similar product such as Geltrex), which consists of a mixture of extracellular matrix proteins, proteoglycans, and growth factors derived from Engelbreth–Holm–Swarm sarcoma cells, has been extensively used as substrate in hepatic differentiation of stem cells [30,51,52,53,54,55]. However, one known disadvantage of Matrigel is that it suffers from potential batch-to-batch variability resulting in difficulty in generating reproducible cultures [56]. In contrast, synthetic polymers have received considerable attention due to their flexibility in composition and hence tunable properties [57]. In general, the chemical and physical properties of synthetic biomaterials can be controlled by altering the composition and fabrication condition. The following illustrates how biochemical and physical properties of biomaterials have been exploited in supporting hepatic differentiation of stem cells.

### 3.1. Biomaterials Presenting Biochemical Cues

#### 3.1.1. Natural and Composite Biomaterials

Liver ECM is composed of collagenous proteins (different collagen types) as well as non-collagenous proteins and proteoglycans, such as fibronectin, perlecan, lumican and laminin [58]. The spatial expression and distribution of different ECM components has been shown to progressively change between fetal and adult liver [59], and between normal and regenerating liver [60], implicating the potential regulatory role of ECM in hepatic tissue development and regeneration. The principal ECM components in the liver are collagens, in particular collagen type I. Therefore, a number of studies have reported the use of collagen gels or scaffolds for hepatic differentiation of stem cells [31,32,61]. Culturing HLC derived from human iPSC on a 2D collagen vitrogel membrane was shown to promote hepatic differentiation and maturation by reducing the expression of immature hepatic markers (e.g., alpha fetoprotein (AFP)) while increasing the expression of mature hepatocyte markers (e.g., albumin (ALB)) compared with Matrigel coating [61]. Additionally, 3D encapsulation of HLC clumps derived from iPSC in collagen hydrogel could also improve the maturation of HLC when compared with 2D culture using tissue culture dish coated with gelatin [31]. Similarly, encapsulation of embryoid bodies (EB) constructed from ESC in a collagen hydrogel could foster hepatic differentiation by enhancing the expression of hepatic markers such as ALB when compared with seeding the EB on collagen-coated dishes [32]. 

Apart from collagen, other ECM components, including vitronectin and laminin, were also shown to have a beneficial effect on the hepatic differentiation of ESC when used as 2D coating. The vitronectin-coated substrates supported hepatic differentiation at a similar level to Matrigel [33], while certain laminin subtypes outperformed Matrigel as evidenced by a higher expression of CYP450 activity and a shift towards fresh hepatocytes in principal component analysis of 1000 genes [34]. These ECM components could provide a chemically-defined, xeno-free alternative to Matrigel and facilitate good manufacturing practice (GMP) manufacturing of HLC. 

Polysaccharides are another class of natural biomaterial widely used in biomedical research. Alginate encapsulation of MSC improved hepatic differentiation compared with 2D culture by producing HLC with higher albumin and urea secretions [62]. ESC encapsulated in alginate were also shown to be able to differentiate into HLC [63]. Although alginate is considered bioinert and does not provide cell adhesion motifs, it can be modified with peptides, such as Arg-Gly-Asp (RGD), to increase bioactivity [64].

The other natural ECM biomaterial that was investigated is decellularized liver scaffolds [65]. Whole organ decellularization can be achieved by perfusion of the organs with various detergents, and the resulting scaffold maintains the native ECM microarchitecture and retains numerous bioactive signals such as growth factors that are difficult to replicate artificially. Consequently, decellularized scaffolds are a promising biomaterial for tissue engineering [66]. Two independent studies have compared the hepatic differentiation of MSC infused within the 3D decellularized liver scaffold (recellularization) and in 2D culture [67,68]. Both found that greater yields of mature, differentiated HLC were obtained in the scaffold culture, suggesting the possibility to repopulate the decellularized scaffold with stem cells for differentiation and direct implantation. Similar results were demonstrated for differentiation of human liver stem cells into HLC [69]. A direct comparison between decellularized liver scaffold and collagen type I scaffold (poly-L-lactic acid-collagen) demonstrated that the former induced superior hepatocyte maturation of HLC derived from iPSC compared with the latter [70], and that the effect is likely mediated by the complex composition of decellularized ECM compared with the single ECM component in the collagen scaffold. Furthermore, when used as a 2D coating, decellularized ECM was shown to outperform collagen, fibronectin and Matrigel in enhancing the hepatic differentiation of MSC [71,72]. Despite the promise it shows, batch-to-batch variability of scaffold properties could limit the widespread use of decellularized scaffolds for directing stem cell differentiation [73]. 

Composite biomaterials with two or more natural polymers have also been reported. The conjugation of heparin, which has a high affinity for a variety of growth factors, to a collagen scaffold led to enhanced hepatic differentiation of MSC as indicated by higher percentage of cells expressing cytokeratin 19 and ALB [74]. It is likely that heparin immobilizes growth factors such as hepatocyte growth factor, a known growth factor for hepatocyte differentiation, to present the signals locally. In another study, three different composite scaffolds were constructed including dextran-gelatin, chitosan-hyaluronic acid, and gelatin-vinyl acetate [75]. Based on the assessment of hepatic marker expression and metabolic functions, all 3D scaffolds outperformed 2D culture and gelatin-vinyl acetate was found to be the most preferable scaffold to support differentiation of MSC into HLC. The authors attributed their findings to the fact that collagen, where gelatin is derived from, is a major component of liver ECM. Finally, HepaRG cells, which is a human HPC cell line and shares features with oval cells such as being bipotent, were cultured on two composite biomaterials, namely nanofibrillar cellulose and hyaluronan-gelatin hydrogels [76]. Both hydrogels induced the formation of HepaRG spheroids and stimulated the hepatic differentiation *via* increasing hepatic marker expression and metabolic functions. Since the difference of the differentiation efficiencies was small between the two biomaterials, the induction effects are likely due to the restoration of cell–cell interaction in spheroid culture rather than specific ECM cues.

#### 3.1.2. Synthetic Biomaterials

The advantages of synthetic biomaterials include tunable and reproducible properties, such as stiffness, degradation rate, swelling rate. Synthetic biomaterials such as poly(lactic-co-glycolic acid) (PLGA) and polyethylene glycol (PEG) are often used to fabricate 3D scaffolds [77]. In order to present biochemical cues close to those in the *in vivo* environment, composite biomaterials consisting of both synthetic and natural polymers with physical or chemical interactions have been developed. For instance, a collagen-coated PLGA 3D scaffold was used to support hepatic differentiation of MSC [78], with the expression of hepatic markers appearing earlier and the metabolic functions higher compared with 2D culture. In another report, a PEG-based scaffold constructed with an inverted colloidal crystal approach and containing uniform pores (100 μm in diameter) was coated with fibronectin and/or collagen [79]. Results showed that the presence of both biochemical cues produced superior induction effects on the hepatic differentiation of MSC, when compared with the scaffold coated with collagen only or the 2D control. 

### 3.2. Biomaterial Systems Presenting Physical Cues

When a specific ECM component (e.g., fibronectin) binds with integrins that recognize the motifs on the ECM (e.g., α5β1), focal adhesion kinase is activated to mediate downstream signaling to elicit an ECM-specific response on stem cell differentiation [80,81]. Meanwhile, integrins and the focal adhesion complex proteins are also known as mechanosensors and mechanotransducers that sense and transduce physical/mechanical signals into biochemical signals. Focal adhesions are protein complexes that form upon the binding of integrin to ECM, and link the ECM to the intracellular cytoskeleton. The subsequent interaction between signaling proteins will activate downstream effectors, such as Rho, to influence cell behavior [82]. Examples of physical signals are stiffness and topography. Using a 2D micropatterned heparin hydrogel, Y. Huang et al. showed that soft hydrogels (~400 Pa) promoted MSC differentiation into HLC and hepatic maturation compared with stiffer hydrogels (up to ~43 kPa) [83]. Similarly, collagen-coated polyacrylamide hydrogels with stiffness of 20–140 kPa were shown to enhance albumin secretion as well as the metabolic activities of HLC derived from ESC compared with tissue culture dish (~3 GPa) [84]. Moreover, culturing resident liver stem cells on soft polyacrylamide hydrogels (400 Pa) enhanced the differentiation of resident liver stem cells to hepatocytes compared to stiffer hydrogels (80 kPa) [85]. The above findings are consistent with the observation that primary hepatocytes cultured on substrates with stiffness similar to that of the liver (∼10 kPa) maintain their differentiated phenotype for longer durations [86]. For 3D culture, encapsulating HepRG cells in a PEG-hyaluronic acid hydrogel conjugated with hydrolytically degradable peptide was used to determine the optimal stiffness environment for differentiation [87]. The elastic modulus of each hydrogel was modulated dynamically due to the combined effects of hydrogel degradation and extracellular matrix production by the encapsulated cells. At a stiffness (2.8 to 6.17 kPa) close to that of the native liver, hepatic differentiation was more mature in terms of hepatic gene expression, albumin secretion, CYP3A4 activity, and drug metabolism. 

Other physical properties which highly influence 3D culture efficiency are porosity and pore size. A polymer scaffold with high porosity, such as an electrospun fibrous scaffold, facilitates metabolic functions of hepatocytes and allows exchange of nutrients and wastes [88]. A porous poly(L-lactic acid)-co-poly (ε-caprolactone) (PLACL)/collagen scaffold with 89% porosity was found to enhance MSC differentiation into HLC compared with PLACL and collagen scaffolds [89]. Scaffolds with large surface to volume ratios have also been shown to promote hepatocyte attachment [90]. In terms of pore size, collagen scaffolds fabricated with various pore sizes (10, 18, 82 µm) were shown to influence the secretory function of hepatocytes and their cell–cell interaction [91]. Specifically, a scaffold with 18 µm pores was found to reduce hepatocyte secretory function as well as expression of cell–cell adhesion proteins compared with scaffolds with 10 µm or 82 µm pores. This was attributed to the significantly higher degree of cell spreading within the 18 µm scaffold which repressed hepatic differentiation. 

In order to present topographical cues, nanoscale fibers can be fabricated to mimic the native ECM architecture [92]. Signal transduction can be triggered when stem cells are exposed to cues *via* integrin binding/focal adhesion formation. Cells exposed to topographical signals have been shown to exhibit upregulation of focal adhesion kinase expression as well as phosphorylation. This is expected to trigger downstream signaling as described in the previous paragraph [93]. In one example, topographical cues presented by electrospun nanofibers (chitosan/polycaprolactone) were shown to influence stem cell differentiation [94]. When ESC were cultured on the fibers of 400 nm and 1.1 µm in diameter, ectodermal commitment was stimulated whereas fibers of 200 nm in diameter promoted hepatic differentiation. A nanopatterned surface with 120 nm of pit spacing was fabricated by electron beam lithography and was shown to promote hepatic differentiation of HepaRG compared with tissue culture dish [95]. These results strongly suggest that tailored ECM-like substrates are capable of influencing the hepatic differentiation of stem cells.

## 4. Comparison between 2D and 3D Biomaterial Systems 

It has been increasingly recognized that 3D cell culture provides a more realistic biochemical and physical microenvironment than 2D cell culture [96,97]. In directing stem cell differentiation, 3D cultures such as generating embryoid bodies or spheroids, which mimic the early stages of embryogenesis and morphogenesis, have been shown to improve differentiation efficiency when compared with 2D culture [98]. In addition, embedding stem cells in 3D scaffold was also seen to enhance the efficiency of stem cell differentiation [99]. Interestingly, the 2D and 3D configurations of stem cells could lead to differential responses when cells are exposed to the same ECM cue. Here we summarize the studies of hepatic differentiation of stem cells in 2D and 3D biomaterial systems with the aim of elucidating the biomaterial effect in different cell culture configurations (Table 1). Although there is a lack of studies comparing 2D and 3D biomaterial systems directly, we can still note the following observations. 1. Decellularized ECM and collagen, the major ECM component in liver, can support hepatic differentiation in both 2D and 3D configurations. In general, encapsulating stem cells in a collagen scaffold produced more mature HLC than inducing stem cell differentiation on a collagen/gelatin-coated substrate [31,32]. 2. Soft biomaterial (< 10 kPa) enhanced hepatic differentiation in both 2D and 3D configurations [84,87]. 3. Whereas biomaterial is always introduced at the beginning in 2D differentiation, the timing of embedding stem cells in biomaterial for 3D differentiation could vary. Biomaterials could either be supplied to guide lineage specification at the initial stage [78], or be supplemented after HLC were generated to boost their maturation [31]. This suggests the ECM cues may have a temporal effect which has not yet been revealed. A systematic investigation should be conducted to study the temporal effect of ECM cues by introducing a biomaterial scaffold to stem cell differentiation at various time points. Meanwhile, the mechanistic knowledge of how 2D and 3D biomaterial systems may differentially regulate stem cell differentiation is limited. One study reported that integrin expression was altered when ECM cues were presented in 2D and 3D [100]. This is likely to affect the expression of downstream signaling molecules, such as focal adhesion kinase (FAK) and extracellular signal-regulated kinase (ERK) [80], which warrants further investigation.

## 5. Conclusions and Future Perspectives 

The capability of stem cells to differentiate into HLC is well documented, but producing HLC with a mature hepatocyte phenotype remains a challenge. Previously, functional maturation of HLC was observed by comparing the phenotype of HLC before and after implantation into an animal model [17]. This implied that maturing stimuli are present in the *in vivo* cellular environment. Notwithstanding, traditional approaches to induce stem cell differentiation rely primarily on soluble chemical signals only. In addition to soluble factors, the cellular environment in the liver also comprises supporting cells and ECM in direct contact with hepatocytes. To recapitulate the ECM environment, both natural and synthetic materials can be used to provide biochemical and physical cues. While Matrigel and decellularized ECM can recapitulate the complex compositions of native ECM, the increasing use of chemically-defined biomaterials should facilitate the design of biomaterials to promote hepatic differentiation of stem cells as well as GMP production of HLC for clinical applications. For example, an optimized scaffold can be fabricated based on a combination of chemically-defined biomaterials, such as laminin and vitronectin, which have been shown to enhance hepatic differentiation and maturation in previous studies [33,34]. As mentioned in the beginning of the article, the crosstalk between different hepatic cell types is important in promoting hepatic differentiation. Given that different hepatic cell types are located at specialized locations in the liver sinusoid and the distribution of liver ECM is heterogeneous [101], the biomaterial scaffold could also be optimized by providing the spatial organization of various cell types as well as ECM components to allow the fabrication of a fully functional artificial liver. 

In this review article, we have provided an overview of various 2D and 3D biomaterial systems applied in hepatic differentiation of stem cells. Although the similarities and differences between the effect of 2D and 3D biomaterial systems were not covered in great detail due to a lack of relevant studies, we have shown that the differences between 2D and 3D biomaterial systems could potentially impact the differentiation efficiency. Therefore, efforts should be devoted to further elucidating how biomaterial configurations influence hepatic differentiation of stem cells and optimizing the differentiation protocol in terms of biomaterial composition and configuration. The biomaterial effect on hepatic differentiation of stem cells mediated by overexpression of microRNAs (miRNAs) such as miRNA-122 and miRNA-194 as well as overexpression of transcription factors such as Foxa2/Hnf4a should also be investigated [102,103,104]. We envision that the combined and optimized use of soluble factors and biomaterial scaffolds should pave the way for more efficient derivation of useful HLC in the future.

## Figures and Tables

**Figure 1 bioengineering-07-00047-f001:**
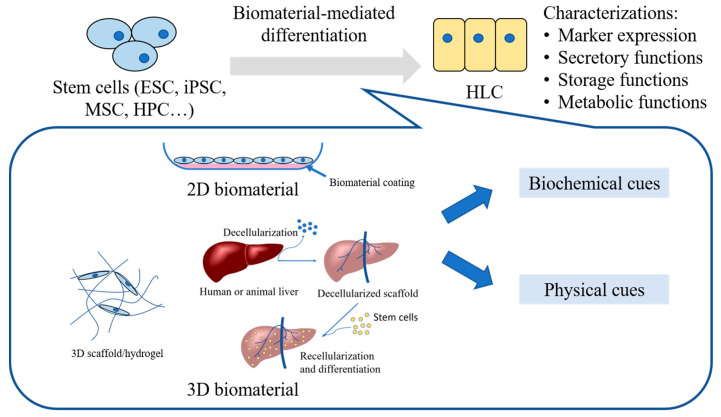
Schematic diagram of 2D and 3D biomaterial-mediated hepatic differentiation of stem cells.

**Table 1 bioengineering-07-00047-t001:** Summary of 2D and 3D biomaterial systems applied in hepatic differentiation of stem cells.

Biomaterial Systems	Stem Cell Sources	Differentiation Efficiency (% of Albumin-Positive Cells)	Ref.
**Biochemical cues**			
2D			
Collagen	iPSC	54.3% (day 25)	[61]
Decellularized liver ECM	MSC	26.7% (day 21)	[71]
Laminin	ESC	91.3% (day 18)	[34]
Matrigel	ESC, iPSC, MSC	80.9% (day 20) [30]; 90% (day 17) [52]; 91% (day 14) [53]	[30,51,52,53,54,55]
Vitronectin	ESC, iPSC	Not provided	[33]
3D			
Alginate	ESC, MSC	87% (day 20) [63]	[62,63]
Collagen	ESC	Not provided	[31,32]
Cellulose, hyaluronan-gelatin	HepaRG	Not provided	[76]
Decellularized liver ECM	ESC, iPSC, MSC	Not provided	[67,68,70]
Dextran-gelatin, chitosan-hyaluronic acid, gelatin-vinyl acetate	MSC	Dextran-gelatin: 57.2% (day 28); chitosan-hyaluronic acid: 62.8% (day 28); gelatin-vinyl acetate: 68.1% (day 28)	[75]
Heparin-collagen	MSC	Not provided	[74]
PEG-collagen/fibronectin	MSC	Not provided	[79]
PLGA-collagen	MSC	Not provided	[78]
**Physical cues (stiffness/topography/porosity and pore size)**			
2D			
Heparin (stiffness)	MSC	~60% (day 21)	[83]
Polyacrylamide (stiffness)	ESC, iPSC, Resident liver stem cells	Not provided	[84,85]
3D			
Chitosan (topography)	ESC	Not provided	[94]
PEG/hyaluronic acid (stiffness)	HepaRG	Not provided	[87]
Poly(L-lactic acid)-co-poly (ε-caprolactone) (PLACL)/collagen (porosity)	MSC	Not provided	[89]

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
