# Peer review of "Hepatic Differentiation of Stem Cells in 2D and 3D Biomaterial Systems"

_bioengineering, 2020, doi:10.3390/bioengineering7020047_

Round 1
Reviewer 1 Report
Review of manuscript: “Hepatic differentiation of stem cells in 2D and 3D biomaterial systems” Bioengineering (ISSN 2306-5354)
This paper deals with an interesting topic, which may potentially contribute to filling some gaps in the existing literature in the area of tissue engineering, however I have a number of minor concerns regarding the review discussion and presentation.
Minor comments:
- The introduction should conclude clear subsection showing the key messages and significance of the contribution of this review study. Also, please add the paragraph, which clearly describes article focus and specifies the aim of this paper. In the current form, the introduction section on p. 2 only indicates that this paper will “discuss the application of various biomaterial systems in hepatic differentiation of stem cells 77 in 2D and 3D culture (…) highlighting their role in promoting lineage specification and hepatic 78 maturation”. Therefore, this statement only says what the authors are doing, but it does not explain sufficiently the aim of this paper, key messages etc.
2.More discussions about biomaterials should be added to this paper.
The paper would be stronger, if it discussed in addition the results from the biomaterials not only in the conventional but also in more advanced processing technologies. It would be beneficial if the authors could add specifically what other studies have found about the cell‐sheets process, 3D bioprinting, and microfluidic systems in the context of specific biomaterial systems, as well as what are the future trends in these major fields.
It would be also helpful and interesting to discuss how the application of small molecules and micro-RNAs or genetic manipulation favours hepatic differentiation of distinct stem cells in the context of specific biomaterial systems.
Please write also a short discussion how the knowledge from this paper can potentially contribute to better the design of early trials to improve the development of an engineered tissue construct, which are based on seeding cells onto biomaterial scaffolds.
- In conclusions section, it should be highlighted what are the strength and the weaknesses of this review.
- The references could be also updated (they do not include the resent publications as there is only only one from 2020 and six from 2019), because some new publications in this area are available. Consider to re-search the literature and include more recent literature, like for example the following papers:
Alain da Silva Morais ,Sílvia Vieira ,Xinlian Zhao ,Zhengwei Mao , Changyou Gao , Joaquim M. Oliveira , Rui L. Reis. Advanced Biomaterials and Processing Methods for Liver Regeneration: State‐of‐the‐Art and Future Trends.(2020)
Katie Morgan, Anna Bryans, Filip Brzeszczynski, Kay Samuel, Philipp Treskes, Joanna Brzeszczynska, Steve Morley, Peter Hayes, Nicolaj Gadegaard, Lenny Nelson, and John Plevris, Oxygen plasma substrate and specific nanopattern promote early differentiation of HepaRG progenitors. (2020)
- The paper is quite well written, although there are some writing errors in some places , for example:
- on p. 2, line 84 use italic font for “in vitro”.
- line 55, 73, 89, 192, 219 use italic font for “via”
Reviewer 2 Report
General comments:
This review article describes the role of current 2D and 3D culture systems in hepatic differentiation of different stem cells sources for therapeutic applications as an alternative to liver transplantations.
Although the review is focused on an important and emerging area of regenerative medicine, it lacks clarity, details (section 2 and 4 are underdeveloped), and proper citations (original articles should be cited).
The authors suggest that the temporal use of biomaterials may improve differentiation efficiency (line 244-245). However, the authors reference current use of stepwise differentiation into HLCs in both 2D and 3D culture (ref [8], line 56) in the introduction, but these techniques are not described in their review.
The manuscript could be improved if current differentiation procedures, characterization of derivatives, and criteria for cell functionality were clearly explained in Section 2, and this detail was added to Figure 1.
Furthermore, Table 1 does not supplement the text, perhaps inclusion of outcomes/ differentiation efficiency would better summarize stem cell differentiation in 2D and 3D culture systems.
Specific comments:
Line 71-72: “Physical properties … differentiation.” should be cited.
Line 88- 89: “Both ESC and iPSC… [20].” Only one study has shown this? Studies should be properly cited.
Line 125-226: Therefore, a number of studies have reported…” What studies? They should be cited here.
Line 139-140: “outperformed Matrigel based on analysis of metabolic functions and genome-wide analysis [44].” Should expand on what the findings were and the importance of these outcomes.
Line 155-157: Should be mentioned earlier in the paragraph when 2D culture is mentioned.
Line 160-165: Paragraph starting from “Polysaccharides… [56]”. Should be described earlier, prior to decellularized liver scaffolds, or be more clearly presented. Overall, organization of Section 3.1.1. could be improved for clarity.
Line 196-225. Organization of Section 3.1.2 describes synthetic and then natural composite biomaterials; however, Section 3.1.1 describes natural materials. This section should be re-organized for readability.
Line 200-202 and Line 219-220: Statements regarding the role of “Signal transduction… via integrin binding/focal adhesion formation [72].” need further justification and discussion.
Line 234: “a previous study… vascular smooth muscle [78]” This citation is not relevant.
Reviewer 3 Report
In this review Zhao et al. summarized and compared various 2d and 3d culture systems challenging stem cell differentiation toward functional hepatocytes. The state of art in the field has been here documented and discussed enough. However, this reviewer suggests minor points that, if addressed, could improve the quality of the manuscript.
1) It would be advisable to cite fewer reviews, to avoid this manuscript becoming a review of reviews. The authors should rather pinpoint the original key works and data that contributed to the advancement of knowledge and technologies in the field.
2) An aspect that the authors should address, at least in the introduction, is the importance that cross talk between the various hepatic cell types plays in the differentiation of precursor cells and in the maintenance of fully functional hepatocytes. This is relevant in the design of bio-scaffold for 3d culture that has to take in account the spatial organization and reciprocal interactions of more than one cell type (see e.g. : Pettinato et al. “Generation of fully functional hepatocyte-like organoids from human induced pluripotent stem cells mixed with Endothelial Cells” Scientific Reports volume 9, Article number: 8920 (2019).
3) A physical characteristic of the bio-scaffold highly influencing the 3D culture efficiency and that was neglected by the authors is the porosity. This is an important physical characteristic of the biomaterial used in 3d culture and particularly for the liver bioengineering, where porosity has been shown to influence not only the exchange of nutrients and waste materials, but also the polarity of the cells and the correct cell-cell interactions. At this regard we suggest to discuss and introduce at least the following references:
Rajendran, D.; Hussain, A.; Yip, D.; Parekh, A.; Shrirao, A.; Cho, C.H. Long-term liver-specific functions of hepatocytes in electrospun chitosan nanofiber scaffolds coated with fibronectin. J. Biomed. Mater. Res. A 2017, 105, 2119–2128.
Ranucci, C.S.; Kumar, A.; Batra, S.P.; Moghe, P.V. Control of hepatocyte function on collagen foams: Sizing matrix pores toward selective induction of 2-D and 3-D cellular morphogenesis. Biomaterials 2000, 21, 783–793.
Round 2
Reviewer 2 Report
The authors addressed majority of the concerns and has appropriately revised the manuscript.